# Characteristics of Adverse Events in Bee Venom Therapy Reported in South Korea: A Survey Study

**DOI:** 10.3390/toxins14010018

**Published:** 2021-12-27

**Authors:** Kyeonghan Kim, Hyein Jeong, Gihyun Lee, Soobin Jang, Taehan Yook

**Affiliations:** 1Department of Preventive Medicine, College of Korean Medicine, Woosuk University, Wanju 55338, Korea; solip922@woosuk.ac.kr; 2Department of Preventive Medicine, College of Korean Medicine, Kyung Hee University, Seoul 02447, Korea; frogcream@khu.ac.kr; 3Department of the History of Korean Medicine, College of Korean Medicine, Dongshin University, Naju 58245, Korea; glee@khu.ac.kr; 4Department of Preventive Medicine, College of Korean Medicine, Daegu Haani University, Gyeongsan 38610, Korea; Soobin@dhu.ac.kr; 5Department of Acupuncture & Moxibustion Medicine, College of Korean Medicine, Woosuk University Hospital of Korea Medicine, Jeonju 54986, Korea

**Keywords:** bee venom, bee venom therapy, bee venom acupuncture, safety, survey

## Abstract

This study was aimed at investigating Korean patients’ experience with bee venom therapy (BVT) and providing evidence to enhance BVT safety. Thus, an anonymous online survey was conducted between August 22 and 28, 2018. Five hundred respondents who underwent pharmacopuncture (PA) within one year were surveyed (sample error: 95 ± 4.38%). Of these, 32 respondents were excluded and 468 were evaluated. Of the 468, 61 reported experiencing adverse events after PA. The adverse event rate was higher in the BV-PA(Bee venom-Pharmacopuncture) group than in the non-A group; however, intergroup differences were insignificant. There were no significant differences in mild symptom intensity between the BV-PA and non-BV-PA groups (*p* = 0.572). However, there was a significant intergroup difference in severe symptom intensity (*p* < 0.001). Additionally, the BV-PA and non-BV-PA groups did not significantly differ in their level of satisfaction either overall or in terms of effectiveness and safety (*p* = 0.414, *p* = 0.339, and *p* = 0.675, respectively). Furthermore, the BV-PA and non-BV-PA groups did not differ regarding intent to re-treat (*p* = 0.722). Severe adverse events such as anaphylactic shock were not reported; however, BVT practitioners should be cautious when applying it.

## 1. Introduction

Bee venom (BV) is one of the most widely-used animal venoms, and consists of various complex compounds that induce allergic reactions [1]. Bee Venom therapy (BVT), in which bee venom is used for medicinal treatment, is applied worldwide; however, it is mainly used in Asia, Eastern Europe, and South America [2]. Particularly in East Asian countries, including Korea, BV is mainly used in pharmacopuncture (PA), which is a traditional medical treatment that combines acupuncture and herbal medicine, unlike traditional acupuncture [3]. In a survey conducted on the general public, 69% of all citizens said they had experience using Korean medicine. In the case of treatment, the PA was 24.1%, similar to that of the herbal medicine (26.8%) [4]. Eighty-eight percent of Korean medicine doctors (KMDs) administer PA, 35% of whom use BV, which is diluted to a ratio of less than 1:10,000 to minimize the side effects when as a PA solution [5,6]. BV is a complex compound consisting of several substances, some of which have anti-inflammatory [7], antinociceptive [8], radioprotective [9], antimutagenic [10], and anticancer [11,12] activities. BVT has shown therapeutic effectiveness against various diseases; in particular, it is known to be effective for inflammatory arthritis and musculoskeletal disease [13,14]. Although the therapeutic effect of BV has been demonstrated through research, there remains an important limitation in that immune responses to BVT can range from trivial skin reactions to life-threatening responses such as anaphylaxis [15,16]. The substances known to be primarily responsible for allergic reactions are phospholipase A2 and histamine, which entail safety concerns for BV-sensitive patients [17,18].

In a recent systematic review (SR), the most common adverse effects of BVT were found to be skin reactions at the injection sites, including pruritus, rash, and swelling, and systemic symptoms such as headache, nasopharyngitis, and pain in an extremity [19]. Research on the safety of BV has been conducted mainly by evaluating toxicity through cell and animal experiments. South Korea has a long history of BV-PA use in traditional Korean medicine (TKM) clinics. Therefore, it is advantageous to investigate BVT-associated adverse events in clinical settings. However, until recently, the data on patient-reported adverse events which would be the foundation for enhancing safety have not been available. In this study, we surveyed in detail the characteristics of patients and adverse events associated with BVT as a PA in Korea. The objective was to investigate Korean patients’ experience with BVT and provide research-based evidence to aid in enhancing the safety of BVT.

## 2. Results

### 2.1. Patient Characteristics

There were 468 respondents, including 224 (50.0%) men and 224 (50.0%) women. Table 1 shows the distribution of the participants’ sex, age, monthly income, and education level. The age distribution was as follows: 72 (15.4%), 205 (43.8%), 118 (25.2%), 59 (12.6%), and 14 (3.0%) were aged 20–29, 30–39, 40–49, 50–59, and 60–69 years, respectively. The most common monthly income bracket was 3000–3999 USD (1 USD = 1000 KRW), and most participants (361, 77.1%) had a university degree. The BV-PA and non-BV-PA groups showed significant differences in age; however, there were no significant intergroup differences in sex, monthly income, or education level.

### 2.2. Adverse Events after BV-PA

Of the 468 respondents, 61 (13.0%) responded that they had experienced adverse events after PA. The BV-PA group (16.7%) had a higher rate of adverse events than did the non-BV-PA group (11.6%); however, the intergroup difference was not significant. Adverse events were classified as mild or severe based on whether additional treatment was necessary. Adverse events included point pain, redness, swelling, and numbness resolving within 24 h without additional treatment; however, some symptoms, including headache, hyperventilation, and suffocating chest pain disappeared after TKM treatment. The intensity of both mild and severe symptoms was measured on a 5-point Likert scale (1, very mild; 2, mild; 3, moderate; 4, severe; and 5, very severe). There were no significant differences between the BV-PA and non-BV-PA groups in terms of mild symptoms (*p* = 0.572); however, there was a significant difference between the groups in severe symptoms (*p* < 0.001) (Table 2).

### 2.3. Satisfaction of BV-PA

Satisfaction and intent to seek re-treatment were measured on a 5-point Likert scale (1, very unsatisfied/definitely would not; 2, unsatisfied/probably would not; 3, neutral/might; 4, satisfied/probably would; and 5, very satisfied/definitely would). The BV-PA and non-BV-PA groups showed no significant differences in overall satisfaction (3.42 ± 0.8 vs. 3.36 ± 0.79, *p* = 0.414), effectiveness satisfaction (3.57 ± 0.98 vs. 3.48 ± 0.87, *p* = 0.339), and safety satisfaction (3.12 ± 0.86 vs. 3.16 ± 0.84, *p* = 0.675). In addition, there was no difference between the BV-PA and non-BV-PA groups in terms of the intent to seek re-treatment (3.45 ± 0.83 vs. 3.43 ± 0.78, *p* = 0.722) (Table 3).

## 3. Discussion

BV-PA is a therapy in which venom extracted from the poison sac and then processed is injected into the human body at specific acupuncture points [20]. The most commonly-used PA solutions are bee venom, ginseng, and hominis placenta [5]. Since experiments in mice have shown that BV can have toxic effects on the liver and kidneys, damage the adrenal glands, and be a strong irritant, care should be taken in clinical applications [21]. In previous studies [22,23], more than half of the reported PA cases with adverse reactions were attributable to BV. Nevertheless, owing to the clinical usefulness of BV, methods that can be used safely are continuously studied. In Korea, BVT is mainly administered in the form of BV-PA [23]. In this study, we evaluated the differences between the BV-PA and non-BV-PA groups in terms of demographic characteristics, adverse events experienced, and treatment satisfaction.

We found that a total of 468 respondents were treated with PA, including 132 (28.2%) treated with BV-PA and 336 (71.8%) without (non-BV-PA). There were no significant differences in sex, monthly income, and education level between the two groups, but there were significant differences in age. In those in their 40s and 50s, BV-PA was more commonly used compared to non-BV-PA. BV-PA is usually used for treating chronic degenerative diseases such as musculoskeletal system diseases in TKM clinics. Therefore, it could be estimated that BV-PA was more commonly used in middle-aged people. However, BV-PA has a relatively strong intensity, and it is difficult to use it in elderly patients [20]. These findings suggest that considering the effectiveness and adverse events of BVT, the appropriate age groups for its use could be individuals in their 40s and 50s. 

The adverse event rate was higher in the BV-PA group (16.7%) than in the non-BV-PA group (11.6%); however, there was no significant intergroup difference. Mild symptoms such as point pain, redness, swelling, and numbness were reported by both groups, and there were no significant differences in the intensity of the symptoms between the groups. In addition, these symptoms disappeared within 24 h without additional treatment. Symptoms were classified as severe if additional treatment was required to manage adverse events. Hyperventilation and suffocating chest pain were reported as severe symptoms by the BV-PA group, and headache was reported by the non-BV-PA group; the intensity of the severe symptoms in the BV-PA group was significantly higher. Even if severe symptoms developed, they alleviated within an hour after the patient rested comfortably in the clinic, drank warm water, or took painkillers. There were no reports of infections after PA, and emergency medications such as epinephrine and dexamethasone were not administered. A negative venom skin test was not always a guarantee of safety [24], and it was estimated that no markedly severe symptoms such as anaphylactic shock developed among the more than 100 BV-PA-treated patients because TKM doctors generally conduct skin tests before treatment [5]. Adverse events related to bee venom therapy are still controversial. According to a recent meta-analysis [25], bee venom acupuncture increased the relative risk of adverse events compared to normal saline. In another review [26], on the other hand, only a few serious cases of bee venom were reported, and these were not considered a cause for concern considering its potential to treat many diseases. 

In a study [5] investigating satisfaction with PA among TKM doctors on a 5-point Likert scale, effectiveness satisfaction was 4.01 and safety satisfaction was 3.66. In this study, effectiveness satisfaction was 3.50 and safety satisfaction was 3.15. Medical costs are a major factor influencing patient satisfaction, and the difference in satisfaction between TKM doctors and patients could be linked to the cost of PA [27]. PA is not included in the national health insurance in Korea; hence, patients have to pay two to five times the cost of general acupuncture treatment. The general public chose PA (33.5%) as the fourth-most expensive oriental medicine treatment, after herbal medicine (68.6%), herbal preparation (58.5%), and Chuna therapy (36.8%). More than a third (33.4%) of the general public also responded that the coverage of insurance benefits should be expanded when asked about priority improvements. People who were experienced with Korean medicine (outpatient 51.6%; hospitalization 67.1%) were even more likely to provide this response, [4].

More than a third (33.4%) of the general public also responded that the coverage of insurance benefits should be expanded in “priority improvements”.

In the case of users of oriental medical, outpatient (51.6%) and hospital patient (67.1%) groups gave the same response. In this study, satisfaction and intent to seek retreatment did not significantly differ between the BV-PA and non-BV-PA groups. This could be acceptable to BV-PA for patients, considering that compared to non-BV-PA, BV-PA is associated with more complicated procedures such as skin tests [6], severe pain [16], high costs [5], and the possibility of adverse events [23].

In Korea, PA including BV-PA is regarded as injection which must be sterilized. TKM doctors mainly bought PA in external herbal dispensaries (EHD). The Ministry of Health and Welfare initiated an EHD certificate system in March 2018. By this standard, EHD facilities should be equipped in keeping with the to Korean Good Manufacturing Practices (KGMP). In November 2021, there were three certified manufacturers, Jaseng, Namsangcheon, and Kirin, and most TKM doctors used BV-PA made by these three EHDs [3]. In other words, almost every KM clinic used certified BV-PA to patients. 

There are different types of BV-PA depending on the degree of dilution. Jaseng and Kirin offer three types of BV-PA, while Namsangchen offers one BV-PA, which is not pure but rather mixed with other PA. Indications are mainly focused on musculoskeletal disorders or inflammation. TKM doctors usually inject into an ashi point, the specific area where the patients complain of pain (Table 4).

According to research about BVT for articular disease in the Journal of Korean Medicine, the most frequently used concentrations were 1:10,000 and 1:3000. Including sweet bee venom, which is composed of pure melittin only, most KMD used concentrations between 1:10,000 and 1:3000. There was no particular pattern in the dilution and type of BVT for each joint outside this trend [28].

In order to safely use BV-PA, there are several requirements which must be complied with. First, only qualified personnel may treat patients, using only certified PA. Second, a skin test and post-injection observation in the clinic are necessary in order to manage potential adverse events such as anaphylaxis. Finally, every adverse effect caused by BVT must be recorded, collected and analyzed in order to suitably revise the criteria. Practitioners should additionally be aware of the various adverse events associated with BVT.

This survey study has several limitations. First, this study was based on a retrospective survey, and recall bias may exist. Adverse events are not usual experiences; therefore, patients’ memories may not be accurate. Second, there is a limit to which the results can be generalized, due to convenience sampling. Because we recruited the participants through an online research company, the sample may not be representative of the general population. Lastly, it is difficult to identify the dilution of BV used to treat patients. In Korea, BV is generally diluted 1:10,000; however, the concentration may vary depending on the case.

Nonetheless, this study is meaningful in that no previous surveys have systematically investigated patients’ experiences with BVT. Unlike other consumer surveys, this survey included as participants not only those treated with BV-PA but also those treated without BV-PA, allowing for comparison and increasing the representativeness of the sample. Based on the results of this study, several suggestions can be made to support effective clinical practice and future clinical trials with BVT.

## 4. Conclusions

This study showed Korean patients’ experience with BV-PA compared to other PA. The clinical use of BVT is limited owing to its adverse effects, despite its many clinical advantages. Contrary to concerns, there was no clear evidence to draw a conclusion that BVT is unsafe. BVT treatment should only be carried out after sufficient preparation for potential serious symptoms. Additionally, a reporting system is needed so that adverse events related to bee venom can be recorded and managed.

## 5. Materials and Methods

### 5.1. Study Design and Setting

This was a survey study of patients treated with BVT as a PA in South Korea. The survey was conducted by Gallup Korea, a professional survey research company that manages more than 100,000 online research panels in South Korea. A total of 500 people were surveyed by purposive quota sampling, with 250 people in the metropolitan area and 250 people in the non-capital area. Thirty-two respondents who mistook embedded therapy for PA were excluded from the analysis, and 468 participants were finally selected. The survey was conducted anonymously between August 22 and 28, 2018, with a sample error of 95 ± 4.38%.

### 5.2. Subjects

The company recruited participants considering the following inclusion criteria: (1) age of more than 20 years and less than 70 years, and (2) having experience being treated with PA. The participants were enrolled on a voluntary basis and were informed that they needed to respond to all of the questions on the questionnaire. Types of PA included extracts from raw herbs and animal products such as ginseng, deer antler, astragalus, etc.

### 5.3. Questionnaire

The questionnaire was developed by two BVT experts who discussed and selected items based on a related study [5]. Three TKM doctors who usually administer BV-PA reviewed a draft questionnaire for face reliability as well as readability. Then, a pilot test was conducted that targeted ten patients who had previously been treated with BV-PA. Feedback was collected and we completed the final version of the questionnaire. The final version of questionnaire had 32 questions.

### 5.4. Study Variables

The detailed variables were as follows:(1)Demographic information: sex, age, monthly income, and education level.(2)Treatment experience: types of PA treatment; adverse events after treatment; overall, safety, and effectiveness satisfaction; and intent to seek re-treatment.

Adverse events after treatment were classified into mild and severe; the questionnaire was set as “Have you ever felt point pain, redness, swelling, or numbness after PA?” and “Have you ever felt a nausea, headache, hyperventilation, suffocating chest, or faint after PA?”. This explanation helped survey respondents to respond more clearly.

### 5.5. Statistical Analyses

Frequency analysis was performed for all variables, and values were presented as the mean ± standard deviation. The chi-square test was employed to determine differences in sex, age, income, and education level. The Shapiro–Wilk test was conducted to test for normality, and an independent t-test was used to determine differences between the BV-PA and non-BV-PA groups. Statistical significance was set at *p* < 0.05, and IBM SPSS ver. 18.0 (IBM Co., Armonk, NY, USA) was used for the analysis.

### 5.6. Ethical Considerations

All participants were briefed about the purpose of the study prior to the initiation of the survey. The survey was conducted anonymously. The entire survey process was approved by the institutional review board (IRB) of Woosuk University (IRB number: WSOH IRB H2103-02, 6 March 2021).

## Figures and Tables

**Table 1 toxins-14-00018-t001:** Basic characteristics of respondents.

Demographic Characteristics (*n* = 468)		BV-PA (*n* = 132, 28.2%)	Non BV-PA (*n* = 336, 71.8%)	*p*-Value
Sex	Male	61 (46.2)	173 (51.5)	0.304
	Female	71 (53.8)	163 (48.5)	
Age (years)	20–29	16 (12.1)	56 (16.7)	0.035
	30–39	51 (38.6)	154 (45.8)	
	40–49	36 (27.3)	82 (24.4)	
	50–59	26 (19.7)	33 (9.8)	
	60–69	3 (2.3)	11 (3.3)	
Monthly Income (USD *) * 1 USD = 1000 KRW	Under 990	2 (1.5)	5 (1.5)	0.312
	1000–1999	8 (6.1)	18 (5.4)	
	2000–2999	13 (9.8)	53 (15.8)	
	3000–3999	20 (15.2)	71 (21.1)	
	4000–4999	23 (17.4)	58 (17.3)	
	5000–5999	21 (15.9)	38 (11.3)	
	Over 6000	45 (34.1)	93 (27.7)	
Level of Education	Middle or High school	13 (9.8)	32 (9.5)	0.396
	College	106 (80.3)	255 (75.9)	
	Graduate school	13 (9.8)	49 (14.6)	

* 1 USD = 1000 KRW.

**Table 2 toxins-14-00018-t002:** Adverse Events treated after PA.

Classification		BV-PA (*n* = 132, 28.2%)	Non BV-PA (*n* = 336, 71.8%)	*p*-Value
Have you experienced any adverse events after PA?	Yes	22 (16.7)	39 (11.6)	0.143
	No	110 (83.3)	297 (88.4)	
How intense were the symp-toms? (*n* = 61)	Mild symptoms	12 (54.5)	20 (51.3)	0.532
	Severe symptoms	10 (45.5)	19 (48.7)	
How intense were the symptoms? (5-point Likert scale)	Mild symptoms	2.52 ± 1.14	2.46 ± 1.10	0.572
	Severe symptoms	2.64 ± 2.27	2.10 ± 1.13	<0.001

**Table 3 toxins-14-00018-t003:** Satisfaction of PA.

Classification		BV-PA (*n* = 132, 28.2%)	Non BV-PA (*n* = 336, 71.8%)	*p*-Value
Satisfaction (5-point Likert scale)	Overall	3.42 ± 0.83	3.36 ± 0.79	0.414
	Effectiveness	3.57 ± 0.98	3.48 ± 0.87	0.339
	Safety	3.12 ± 0.86	3.16 ± 0.84	0.675
Intent to re-treatment (5-point Likert scale)		3.45 ± 0.83	3.43 ± 0.78	0.722

**Table 4 toxins-14-00018-t004:** List of approved BV-PA in Korea.

Manufacturer	Classification	Component	Indication	Main Acupoint
Jaseng	B1-BV (5%)	Melittin, Apamin, Phospholipase A2, Hyaluronidase	Musculoskeletal disease, Rheumatoid arthritis, Muscle pain, Tennis elbow, Ankylosing spondylitis	Back-shu point, GV3, GB21, Ashi point
	B2-BV (10%)	Melittin, Apamin, Phospholipase A2, Hyaluronidase		
	B4-eBV	Eliminated phospholipase A2, Histamine		
Namsangchen	CA	Carthamus tinctorius: Cervus nippon:bee venom = 99:1:0.1	Acute or Inflammatory joint disease	Ashi point
Kirin	SBV10	99% pure melittin 0.1 mg/mL	Pain, Inflammatory disease, Auto-immune disease, Musculoskeletal disease	Back-shu point, GV3, GB21, Ashi point
	SBV25	99% pure melittin 0.25 mg/mL		
	SBV50	99% pure melittin 0.5 mg/mL		

## Data Availability

The data presented in this study are available on request from the corresponding author.

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
