# Peer review of "Characteristics of Adverse Events in Bee Venom Therapy Reported in South Korea: A Survey Study"

_toxins, 2021, doi:10.3390/toxins14010018_

Round 1

Reviewer 1 Report

This article describes the survey study results of the adverse events of bee venom therapy reported in South Korea. The study results indicate how often adverse effects occur after using bee venom pharmacopuncture and how severe they are. This topic is significant to face due to the associated risks of the severe adverse reactions that may occur in sensitized people after administering this type of therapy.

This article is quite well written. The obtained results are adequately described, but they should be better discussed with the current literature. However, the research has some limitations (pointed by the Authors).

I have some comments:

Introduction

Line 25: The main allergens contained in bee venom are worth mentioning. Some venom compounds can have pro-inflammatory and some anti-inflammatory properties. There are reports, for example, that phospholipase A2 may have different properties depending on its concentrations. This may be relevant for BV acupuncture.

Line 26: For the readers from outside Korea and Asia, the Authors should provide brief information on which diseases BV therapy and BV acupuncture are most commonly treated. The Authors should also state whether such treatment is effective based on the available literature.

Results

Line 65: I propose to give the percentage of occurrence of mild and severe reactions in patients reporting adverse effects after BV-PA and non-BV-PA

Line 86: The text should point out that these institutions are manufacturers of venom suitable for acupuncture (not clinics). I understand that other clinics in South Korea use these commercial products.

Discussion

Line 131: References style should be standardized.

Line 151: I do not see a close relationship between the safety recommendations for PA use and the results of this study. This study aimed to investigate Korean patients' experience with bee venom therapy (BVT) and provide evidence to enhance BVT safety. It seems that the recommendations given in the Discussion section are general and do not apply specifically to the BV-PA. The Authors should discuss better this point.

Line 162: The study should be completed with data derived from the clinical reports. This article would make more sense if the information on side effects came not from surveys but from examinations carried out in clinics providing PA. Hence, this publication can be regarded as a simple overview study in its present form because it isn't easy to draw reliable conclusions from the presented results. In this case, I think the Authors should better discuss the reports from the available literature concerning the adverse effects of PA depending on the applied method, venom and its concentration.

Materials and Methods

Line 184: What about non-BV-PA patients? What was the PA solution used in this group? Is it reported that the (herbal) substance used in this group can also cause adverse reactions? How can this affect the results of the analyses carried out in this presented study?

Reviewer 2 Report

The article is interesting and valuable. Bee venom known as api-toxin is used in medicine and cosmetology. It is a natural bee product, which has a great therapeutic potential and bring many pharmaceutical advantages. It is also highly effective but possesses an important limitation with side effects. I would like to underline, that the authors paid attention to this aspect. Therefore, conducted survey among large group who underwent bee venom acupuncture and pharmacopuncture provided necessary  information of the side effects during bee venom therapy.

I am under impression of this article. The conducted study was very well planned, but should be better described.

The reviewer suggests major revisions. The list of suggestions and remarks are listed below:

Point 1: In the material and methods section, no information on patient’s inclusion criteria can be found.

Point 2: In the material and methods section, there is no information of the exclusion criteria for patients.

Point 3: In the material and methods section, no information concerning the bee venom source has been implemented.

Point 4: No information on the bee venom therapy standardization in the text of the paper can be found

Point 5: The characteristics of the patients did not consider the body weight and height of the people mentioned in the study

Point 6: In the “Materials and Methods” section the information on Setting standardization protocol has not been included. The authors should describe the conducted survey take into account the number of open and close questions.

Point 7: It is necessary to complete the “Materials and Methods” section with Subjects subsection.

Point 8: Moreover in section 2.2. Authors should precisely describe the amount of people who had experienced adverse effects after bee venom acupuncture and non-bee venom acupuncture. Authors should indicate the number of people with headache or hyperventilation.

Point 9:  On page 5 in line 148 after the decimal point authors should continue word with small letter.

Point 10: The Conclusions section summarize the results. Chapter Conclusions should be modified.

Reviewer 3 Report

The manuscript entitled “Characteristics of the Adverse Events of Bee Venom Therapy 2 Reported in South Korea: A Survey Study” deals with the investigation of Korean patients with BVT, in order to provide research-based evidences to improve the safety of BVT.

The investigation was based in a survey done with 500 patients which reported (or not) to have experienced adverse effects after sections of pharmacopuncture (PA).  As conclusion of the present study, the authors  mentioned:

  • The adverse event rate was higher in the BV-PA group than in the non-A group;
  • intergroup differences were insignificant. Further, there were no significant differences in the mild symptom intensity between the BV-PA and non-BV-PA groups;
  • there was a significant intergroup difference in severe symptom intensity;
  • the BV-PA  and non-BV-PA groups did not significantly differ in overall, effectiveness, and safety satisfaction;
  • the BV-PA and non-BV-PA groups did not differ regarding the intent to retreat severe adverse events such as anaphylactic shock were not reported.

The subject in general is interesting, however the way the authors reported their data sounds strange. There are some serious concerns that must be clarified by the authors, as described bellow:

  1. The survey was responded by patients, not doctors; therefore, it is necessary to clarify some aspects that may be clear for specialists, from those understood by non-specialist. It is necessary to define clearly the criteria to classify the adverse effects  (mild, moderated, and severe) and to state which effects were included in each classification. This aspect must be well defined for the comprehension of non-specialists;
  2. The results of the survey compared adverse effects of BV-PA against non-BV-PA groups. This approach seems to be considering that individuals of “non-BV-PA”  group underwent some types of adverse effects from other PA actions, not related to  BV inoculation. This sounds strange, since it is difficult to compare these groups. Which type of inoculum was used for these individuals ?
  3.  Another point that raise some concern is related to the existence of different companies in South Korea producing inoculum for BV-PA, with different compositions from each other. Was this aspect identified and emphasized in the survey ? Different venom components may cause different adverse effects in the patients, interfering seriously in the results.
  4. Apparently the survey included a series of information not relevant for the objectives of the present investigation, and must be removed from the manuscript.

Reviewer 4 Report

The manuscript describes a survey study of the adverse events of bee venom therapy. It was very impressive to learn about some aspects of bee venom therapy in South Korea and that adverse events of bee venom therapy are rare. As indicated by the authors on table 4, the methods of bee venom therapy in South Korea are not standardized in terms of bee venom quantity, combination of components, and sites for injection, so it will be necessary to conduct a detailed study on individual methods with large sampling scale in the future.

Nevertheless, I support the publication of this manuscript. There are a few minor points that I believe can be improved.

(1) I could not get an overview of bee venom therapy in South Korea. In the introduction, please provide some statistical data such as, how many people receive bee venom annually in South Korea?

(2) I could not agree that cost is a factor in the difference in safety satisfaction between patient and doctor in lines 132-143. Please discuss about this matter in more detail.

Round 2

Reviewer 1 Report

I have no further comments.

Reviewer 2 Report

Dear Authors,

You should use past tense in page 6, line 222

I accept your manuscript

Reviewer 3 Report

The authors have corrected all the points of concerns raised by the reviewers, and accepted all the suggestions to improve the quality of the manuscript. Therefore, the manuscript became recommended for publication in Toxins.